# Albumentations: Fast and Flexible Image Augmentations

**Alexander Buslaev [1], Vladimir I. Iglovikov [2], Eugene Khvedchenya [3], Alex Parinov [4], Mikhail Druzhinin [5] and Alexandr A. Kalinin [6,7,\*]**

1   Mapbox, Minsk 220030, Belarus; al.buslaev@gmail.com
2   Lyft Level 5, Palo Alto, CA 94304, USA; iglovikov@gmail.com
3   ODS.ai, Odessa 65000, Ukraine; ekhvedchenya@gmail.com
4   X5 Retail Group, Moscow 119049, Russia; creafz@gmail.com
5   Simicon, Saint Petersburg 195009, Russia; dipetm@gmail.com
6   Department of Computational Medicine and Bioinformatics, University of Michigan,
    Ann Arbor, MI 48109, USA
7   Shenzhen Research Institute of Big Data, Shenzhen 518172, Guangdong, China
*   Correspondence: akalinin@umich.edu

**Abstract:** Data augmentation is a commonly used technique for increasing both the size and the diversity of labeled training sets by leveraging input transformations that preserve corresponding output labels. In computer vision, image augmentations have become a common implicit regularization technique to combat overfitting in deep learning models and are ubiquitously used to improve performance. While most deep learning frameworks implement basic image transformations, the list is typically limited to some variations of flipping, rotating, scaling, and cropping. Moreover, image processing speed varies in existing image augmentation libraries. We present Albumentations, a fast and flexible open source library for image augmentation with many various image transform operations available that is also an easy-to-use wrapper around other augmentation libraries. We discuss the design principles that drove the implementation of Albumentations and give an overview of the key features and distinct capabilities. Finally, we provide examples of image augmentations for different computer vision tasks and demonstrate that Albumentations is faster than other commonly used image augmentation tools on most image transform operations.

**Keywords:** data augmentation; computer vision; deep learning

## 1. Introduction

Modern machine learning models such as deep artificial neural networks often have a very large number of parameters, which allows them to generalize well when trained on massive amounts of labeled data [1]. In practice, such large labeled datasets are not always available for training, which leads to the elevated risk of overfitting [1–3]. Data augmentation is a commonly used technique for increasing both the size and the diversity of labeled training sets by leveraging input transformations that preserve corresponding output labels. In computer vision, image augmentations have become a common regularization technique to combat overfitting in deep convolutional neural networks and are ubiquitously used to improve performance on various tasks [4–6]. Image augmentations have also been shown to improve convergence [7], generalization and robustness on out-of-distribution samples [8,9], and to overall have more advantages compared to other regularization techniques [10].

While most popular deep learning frameworks such as TensorFlow [11], Keras [12], and PyTorch [13] implement basic image transformations, the range is typically limited to some

variations and combinations of flipping, rotating, scaling, and cropping. Different domains, imaging modalities, and tasks may benefit from a wide range of different degrees and combinations of various image transformations [14–16]. The need to use more sophisticated training set augmentation recipes has been typically addressed by custom implementations of image transformations for the task in hand using low-level libraries, for example OpenCV [17] and Pillow [18]. However, implementing new complex transforms and their combinations from scratch can be challenging, time-consuming, and error-prone [19], especially in tasks with complex targets such as image segmentation, as we discuss further in Section 2.3. The development of the image augmentation-specific tools, for instance imgaug [20], Augmentor [21], and CLoDSA [22], aimed to fill in this gap. However, existing solutions typically focus on a variety of operations, processing speed, or flexibility of the application programming interface (API), at the cost of other factors. Thus, there is a need for a flexible image augmentation tool that allows combining a wide range of image transforms and annotation types.

In this paper, we present Albumentations, an open source Python library for fast and flexible image augmentations: https://github.com/albumentations-team/albumentations. Albumentations efficiently implements a rich variety of image transform operations that are optimized for performance, and does so while providing a concise, yet powerful image augmentation interface for different computer vision tasks, including object classification, segmentation, and detection. We demonstrate that Albumentations is faster than other popular image augmentation tools on most commonly used image transformations, without sacrificing the variety of operations or an ability to compose them into more complex pre-processing pipelines.

## 2. Background

### 2.1. Approaches to Image Augmentations

Recent successes of deep learning are often attributed to the advances in the development and availability of algorithms, hardware, and large labeled datasets. In computer vision, ImageNet database and the Large Scale Visual Recognition Challenge [23] have become the main platforms for benchmarking modern deep learning models for image classification after the breakthrough performance of AlexNet in 2012 that nearly halved previous state-of-the-art error rates [24]. Basic image augmentations have been identified as an essential technique for training neural networks for visual recognition before that [25], and AlexNet extended their use to random crops, translations, horizontal flips, and altering the intensities of RGB channels to achieve that seminal result.

After AlexNet, multiple studies demonstrated the effectiveness of more "aggressive" image augmentation strategies that extended image crops with extra pixels and added additional color manipulations [26–28]. Very extensive image augmentations have become commonly employed in medical image analysis, where datasets are often small and expensive to acquire, and annotations are often sparse, compared to the natural images [29–31]. More recently, multiple novel approaches to augment input images were proposed, e.g. Cutout [32], Mixup [33], their derivatives [34] and combinations [35]. Most of these approaches used a fixed set of image transforms and suggested that further improvements can likely be made by simply expanding the pool of used data augmentations.

However, it has also been reported that redundant or overly aggressive augmentation can hurt the performance and introduce biases into the dataset [36,37]. For example, image rotation is an effective data augmentation method on CIFAR-10, but not on MNIST, where it can negatively affect the network's ability to distinguish between handwritten digits *6* and *9* [15]. Reducing the amount of augmentations in the last few epochs of training was proposed as a way to reduce a distribution gap between clean and augmented data and to improve model's performance [38]. Learning more sensible data augmentations for the specific datasets in hand has been explored [14,15,39]. AutoAugment [16] and Fast AutoAugment [40] used reinforcement learning to find the optimal data augmentation strategies from a discrete search space of transform operations. Population-Based Augmentation [41] focused on generating augmentation policy schedules instead of fixed augmentation recipes.

Although there have been attempts to evaluate individual contributions of image transform operations to overall accuracy [42], different augmentation strategies still lead to performance variability even on the well studied performance benchmarks. Most of the above proposed approaches used a fixed set of image transforms and suggested that further improvements can likely be made by expanding the pool of used image augmentations. This is why we identified the **variety of the available transforms** as one of the core needs for the productive use of image augmentations.

### 2.2. Performance Considerations

It is common practice to execute image augmentations on-the-fly during the model training and not pre-compute them ahead of time. Since the datasets for computer vision tasks are typically large in size, this allows one to overcome limitations of the storage space without sacrificing the diversity and amount of image transformations. Many computations in modern deep learning models have been moved to be executed on general-purpose consumer-grade parallel computing chips, such as graphics processing units (GPUs), that have been getting cheaper over the years. However, image augmentations are typically executed on central processing units (CPUs). While a GPU is processing a batch of images, the next batch is being augmented on a CPU using multiprocessing, such that each image is transformed in a separate process. As GPUs are getting faster, execution of image transforms on a CPU can become a performance bottleneck. Thus, to fully utilize the power of modern GPUs, augmentations on CPUs have to be fast enough and the speed of each transform operation is important.Recently, implementations of GPU-based image augmentations have been presented; however, the full advantage of using these approaches can only be achieved in a setup with a relatively high GPU/CPU core ratio [43].

On the software side, although Python has become a  lingua franca in the machine learning community, it is well known that the Python interpreter is not very efficient for executing computation-heavy algorithms. As a consequence, core components of many Python machine learning libraries are often implemented in a different programming language, such as C++, to achieve better performance. However, the speed of execution of various array operations in low-level libraries can vary significantly. That is, while some operations can be faster in one library, others will be better optimized in another one.

While it is essential for a modern deep learning software to provide a convenient Python interface, it is important to make sure that the performance of an underlying implementation satisfies the needs of researchers and/or developers.

### 2.3. Augmentations of Complex Targets

Many of existing approaches and tools have focused on augmenting images in the image classification setting. However, object detection and image segmentation are also very important tasks that are common, for example, in biomedical image analysis [29,44–47], urban scene understanding [48–50], and other domains [51–53]. Preserving the correct annotation when applying image transforms becomes less trivial in object detection and segmentation tasks due to the need for an adequate transformation of corresponding target as well. This challenge can be further aggravated by the need to augment an image with multiple target annotations, for example, for simultaneous segmentation of object areas and their borders (outlines). Multiple annotations can be used for improving segmentation performance, as was recently demonstrated in the biological image segmentation task by the winning team of the Kaggle 2018 Data Science Bowl [46]. Furthermore, in a multi-task setting, an input image can have multiple targets of different type, for example a segmentation mask and a set of bounding boxes. Since image transforms that are packaged with popular deep learning frameworks typically do not provide augmentations of such complex targets out-of-the-box, there is a need for transforms with the out-of-the-box **support of bounding boxes, segmentation masks, and keypoints**.

Albumentations aims to tackle these challenges by providing a flexible and convenient Python interface for a rich variety of augmentations for image classification, segmentation, and object detection, based on optimized implementations of transform operations that outperform their alternatives.

## 3. Design Principles

Albumentations implements a design that seeks to provide a balanced approach to addressing the existing needs. Overall, it relies on five main design principles.

### 3.1. Performance

In a typical deep learning hardware configuration, CPU can be a performance bottleneck, thus the speed of individual transform operations becomes a top priority. Albumentations strives to deliver the best performance on the most of commonly used augmentations by wrapping multiple low-level image manipulation libraries and choosing the fastest underlying implementation. As a trade-off, Albumentations has to rely on a bigger number of dependencies compared to other high-level wrappers that are based on a single library.

### 3.2. Variety

Since finding an optimal set of augmentations for a particular task, dataset, and domain is still an open research topic, it is important to provide an extensive list of available operations that can be quickly tested for the problem at hand. Therefore, Albumentations aims to implement a very diverse range of image transforms. It includes all or almost all basic and commonly used operations such that most research results from recent studies can be validated on an extended pool of augmentations. It also adds some task- and domain-specific image transformations. This is where the trade-off discussed in Section 3.1 gives Albumentations another advantage, since it now can combine operations that were previously unique to an individual low-level library.

### 3.3. Conciseness

To maximize user productivity and enable quick and easy experimentation with different augmentations and their combinations, Albumentations has to provide a concise and convenient interface that is powerful and intuitively clear. It should provide enough control for fine-tuning parameters of all operations, if needed, but the complexity of transform implementations should be hidden behind the API. With that in mind, we consider augmentations for object detection and image segmentation to be as important as for image classification. Therefore, Albumentations automatically applies transforms to complex target annotations, such as bounding boxes, keypoints, and segmentation masks.

### 3.4. Flexibility

Albumentations is constantly evolving and changing, as new image transforms are being proposed, the community requests support for new features, and the underlying implementations of low-level image operations are being optimized (less frequently). Moreover, we have seen a quick adoption of Albumentations in both Kaggle and research communities, as well as in commercial companies. Contributions from these communities help Albumentations to grow, to test and validate its features, and to define the direction of the future development. Therefore, the architecture of Albumentations should be flexible quickly to adapt to these changes and to enable simple ways to contribute new transforms, parameters, tests, and tutorials.

### 3.5. High Open Source Development Standards

Building open source software enhances the rigor and impact of research by allowing others to reproduce published results [19,54]. For an open source software engineering project to be extensible

and robust, it is also essential for the code base to provide a clear way to make contributions and maintain the high quality of code simultaneously. Albumentations is published under a permissive free software license and has a contribution guide [55]. For each commit to the Albumentations source code, continuous integration tools first perform style check and then run unit tests that cover the entire code base. To the date, we have more than 5000 unit tests that cover standard situations and corner cases that different transforms may encounter.

## 4. Key Features

Over the past few years, multiple image augmentations libraries have been developed, including imgaug [20], torchvision [13], Augmentor [21], CLoDSA [22], SOLT [56], and Automold [57]. Each has its own advantages when used for a specific task/domain/dataset combination. However, these libraries did not satisfy our requirements for a wide enough range of implemented transform operations, performance, or support for multiple targets. Since existing tools lacked a balanced approach, the authors of this paper have been independently developing their own custom solutions to make augmentations execute more quickly, in order to keep GPUs utilization during training close to 100%. These custom implementations in part relied and built upon existing libraries, combining, extending, and modifying available image operations. At some point, these solutions were merged together into what later became Albumentations, with the first public alpha release in June 2018. To support users of different deep learning frameworks, Albumentations provides a convenient Python interface to enable a seamless integration with PyTorch [13], Keras [12], and Tensorflow [11].

Albumentations supports all of the most commonly used image transform operations (see Figure 1) and some domain- or task-specific ones, for example changes in weather conditions for autonomous vehicles perception modeling [57]. The rest of this central section is organized as follows: we list features that make Albumentations stand out compared to other similar solutions, and provide examples of code to illustrate how to use them in practice.

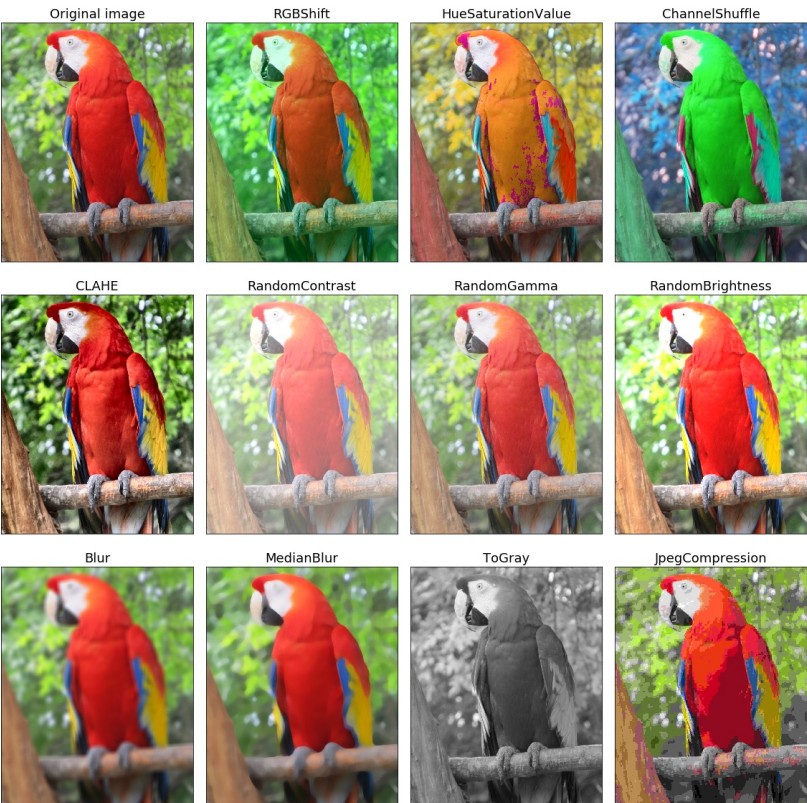

**Figure 1.** Exemplar applications of image transformations available in Albumentations.

### 4.1. Declarative Definition of Parameters

Albumentations follows the best practices of object-oriented design, and each augmentation operation in the library is implemented as a class with clear and documented structure. Class hierarchy diagram is shown in Figure S1. A corresponding class constructor defines a range of random parameters for a specific image transform operation. The example below illustrates parameter definitions:

```
import albumentations as A
aug = A.RandomSizedCrop(min_max_height=(128, 256),
height=224, width=224, p=0.3)
```

In this example, we instantiated an augmentation transform object that crops a portion of an image (the crop size is randomly sampled from the range of $[128, 256]$ pixels) and resizes it to a $224 \times 224$ square image with a 30% chance of that to be actually applied. Probabilistic execution has a great advantage when constructing complex augmentation pipelines. More importantly, declarative definition of parameters for all augmentations makes it very easy to experiment with different combinations of augmentations.

Once created, an instance of any augmentation transform is callable, which allows applying that augmentation to a particular image that is passed as a named argument:

```
image = cv2.imread("test.png")
augmented_dict = aug(image=image)
image_out = augmented_dict["image"]
```

Although Albumentations can be used in the way shown in the example above, one of the design principles is to simplify the use of API as much as possible and reduce the chance of making a mistake. Therefore, Albumentations offers a composition of multiple augmentations in a complex pipeline.

### 4.2. Composition

Composition allows applying multiple augmentations to an input image sequentially or using simple control-flow logic. In a composition, each transformation takes the output of the previous transformation as an input. This simple, yet powerful technique enables building sophisticated pipelines of transforms that in fact can be implemented in different low-level array manipulation libraries. A composition allows to describe such a complex sequence of augmentations in a simple and clear declarative fashion. Albumentations implements a few different ways of composing image transform operators (see Figure S2).

Let us consider a real-life example from one of the top-performing solutions from the APTOS 2019 Blindness Detection Challenge [58]:

```
transform = A.Compose([
A.OneOf([
A.ShiftScaleRotate(..., p=0.5),
A.ElasticTransform(..., p=0.5),
A.OpticalDistortion(..., p=0.5),
A.GridDistortion(..., p=0.5),
A.NoOp()
]),
A.RandomSizedCrop(..., p=0.3),
A.ISONoise(p=0.5),
A.OneOf([
A.RandomBrightnessContrast(..., p=0.5),
A.RandomGamma(..., p=0.5),
A.NoOp()
```

```
    ]),
A.OneOf([
A.FancyPCA(..., p=0.5),
A.RGBShift(..., p=0.5),
A.HueSaturationValue(..., p=0.5),
A.ToGray(p=0.2),
A.NoOp()
    ]),
A.ChannelDropout(p=0.5),
A.RandomGridShuffle(p=0.3),
A.RandomRotate90(p=0.5),
A.Transpose(p=0.5)
])
```

In this example, an augmentation pipeline applies one of spatial augmentations (`ShiftScaleRotate`, `ElasticTransform`, `OpticalDistortion`, `GridDistortion`, or none) as a first step. This type of grid transformations is commonly used in biomedical image analysis (see example in Figure 2). Then, random cropping with resizing occurs with a probability of 30%, followed by a set of color image augmentations (ISO camera noise, brightness, contrast and gamma adjustment, color shift augmentations, and color removal). Finally, a random channel can be dropped out and/or grid shuffle can be applied. In the final step, the image may be randomly rotated by 90 degrees and transposed. This complex augmentation pipeline expressed in a short snippet offers excellent flexibility in trying out multiple augmentation strategies. A more detailed code listing for this experiment is provided in the Supplementary Materials (Listing S11).

The main advantage of this API design is simplicity that allows for a laconic, easily extendable expression of complex workflows.

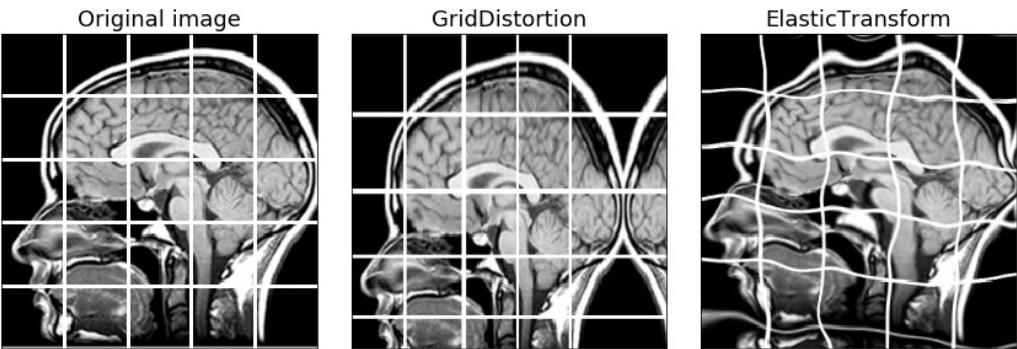

**Figure 2.** Grid distortion and elastic transform applied to a medical image.

### 4.3. Complex Target Support

Image classification is the most common task in computer vision applications of deep learning. Augmenting images in the image classification setting is the easiest, since it does not require applying any transformations to the target value, i.e. an image class. It is less trivial in the object detection or image segmentation and registration tasks, since the corresponding structured image annotations has to be transformed correspondingly. Originally, Albumentations only supported image and segmentation mask augmentations. Due to the growing community demand, bounding box and keypoints augmentations were added later. Each target has its own rules of transformation. While adding bounding box support, we intentionally included support of COCO [59], PascalVOC [60], and YOLO [61] bounding box formats to give users the freedom to choose one of the most commonly used formats that fits their needs best, without adding any additional conversion code. Under the hood, the

internal representation of bounding boxes is normalized to `XYWH`, which allows us to keep sub-pixel coordinates precision in consecutive spatial augmentations.

### 4.3.1. Image and Mask Target Support

Image targets are first-class citizens in the library. It supports augmentation of 8-bit grayscale, RGB, RGBA, 16-bit unsigned integer, and 32-bit floating-point depth images. In addition, it supports multi-spectral image augmentation with a number of channels greater than 4, which is another unique feature of Albumentations. At the moment of the submission, there are 40 pixel and 32 spatial transforms. Any of them could be applied to the input images, as shown in exemplar Figure 1.

In a segmentation task, the same spatial transform, for example a rotation or a crop, should be applied to both input images and target masks (see Figure 3). Moreover, some spatial transformations, such as resizes or rotations, should have enforced nearest-neighbor interpolation for masks to avoid creating nonexistent ghost labels. Let us consider an example task, where one needs to assign class labels 0, 1, 2 to pixels in the image, which correspond to background, pedestrian, and car, respectively. If we take an image that has the only the background and car classes, the target mask will consist of values 0 and 2. However, if we resize and rotate the image and the corresponding mask, using bilinear interpolation, which is the default in all image processing libraries, we may get a mask that will have pixel values of 1. These ghost labels add noise to the training process, and by nature are hard to debug. Albumentations does housekeeping work automatically and all augmentations that support mask augmentation are covered with regression unit tests to verify the mask validity.

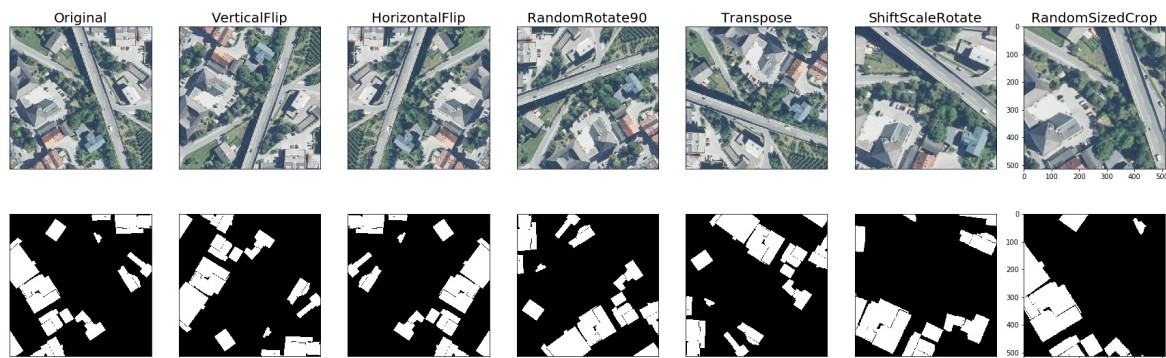

**Figure 3.** An example of geometry-preserving transforms applied to satellite images (**top row**) and ground truth binary masks (**bottom row**) from the Inria Aerial Image Labeling dataset [62].

### 4.3.2. Bounding Box Support

Albumentations supports seamless augmentation of bounding boxes in three most common formats: COCO [59], PascalVOC [60], and YOLO [61]. The library automatically handles the case when bounding boxes go outside of the visible area of an image and allows the user to choose from a few possible options for processing these bounding boxes. To enable bounding boxes augmentation, a top-level `Compose` object should be created with a specific bounding box format passed as an argument:

```
transform = A.Compose([...],
bbox_params=A.BboxParams(format="coco", ...))
...
data = transform(image=original_image,
bboxes=original_bboxes, labels=original_labels)
```

In the code snippet above, we declared that bounding boxes will be in COCO format [59], and bounding boxes with the area less than 128 pixels or with the visibility less than 50% percent after augmentations applied, will be removed.

Some spatial transformations, such as different types of crops, may also change the number of bounding boxes. For example, when we crop the left part of the image, bounding boxes in the right part should be removed. We have a parameter `label_fields` in the definition of a Compose operator in order to keep track of this boxes-labels correspondence.

### 4.3.3. Keypoint Support

Albumentations also supports augmentation of keypoints out-of-the-box. A keypoint defines a location in an image coordinate space with $X$ and $Y$ pixel coordinates (top-left origin) and optional angle ($A$) and scale ($S$) components. One typical use case for keypoint annotations is image registration, a common task in biomedical image analysis [29,63]. In the following example, two key points $X = 10, Y = 20, A = 45, S = 20$ and $X = 34, Y = 40, A = 70, S = 30$ augmented with the `A.ShiftScaleRotate` transform:

```
aug = A.Compose(
[A.ShiftScaleRotate(..., always_apply=True)],
keypoint_params=A.KeypointParams(format="xyas"))
...
aug(image=original_image, keypoints = [[10,20,45,20], [30,40,70,30]])
```

In the case of spatial transformations, these parameters will be updated consistently with an input image. To date, Albumentation does not support flipping of facial landmarks in the case of horizontal flip augmentations. However, additional semantic information about keypoints can be passed as a free parameter to the augmentation transform instance.

### 4.3.4. Multiple Targets

Sometimes, there may be more than one target in a specific computer vision task. They also may be of the the same type or belong to different types, e.g. both segmentation masks and bounding boxes. When augmentations are applied to images with these complex annotations, used transforms need to be translated in separate sets of operations for input images, their masks, and bounding boxes. In the spirit of prividing a powerful, yet concise API, Albumentations supports multiple target management out-of-the-box. The same transform, say rotation by 20 degrees, can be applied to a set of $N$ images, $B$ bounding boxes, $M$ masks, and $K$ keypoints simultaneously. To our knowledge, multiple target support is a unique feature of the Albumentatons library, not supported by other existing solutions. One example with augmenting both segmentation masks and bounding boxes is shown in Figure 4.

### 4.4. Data-Dependent Augmentations

There is a subset of augmentations, for which behavior is dependent on the input data. One example of such augmentation is `MaskDropout`. This augmentation removes some objects from the target mask and zeroes out corresponding pixels in the input image. Another example of input-dependent augmentation is `RandomCropNearBBox`, which randomly crops a part of the image with respect to target bounding boxes, such that each bounding box remains visible in that crop. To the best of our knowledge, these augmentations are also unique to Albumentations.

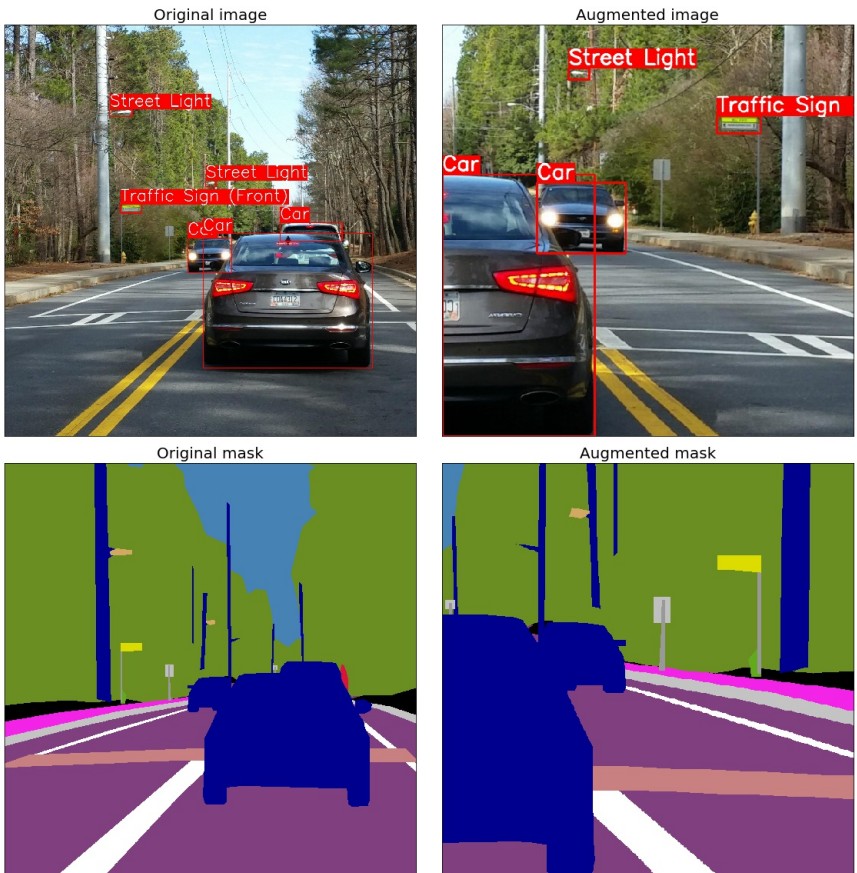

**Figure 4.** An example of applying a combination of transformations available in Albumentations to the original image, bounding boxes, and ground truth masks for instance segmentation.

*4.5. Serialization and Replay Mode*

Deep learning and its applications are often still an experimental science. Many hyperparameters need to be tuned to achieve the best result. To ensure reproducibility of the experiments, researchers typically fix hyperparameters, including those for augmentations, in a configuration file. The standard format options for configuration files are JSON, YAML, and a Python dictionary. Albumentations supports serialization and de-serialization of a `Compose` type object to the file in one of these formats.

```
transform = A.Compose([
A.RandomCrop(...),
A.OneOf([
A.RGBShift(),
A.HueSaturationValue()
]),
])
A.save(transform, '/tmp/transform.json')
```

In the snippet above, we defined an augmentation pipeline and assigned it to the variable `transform`. Then, we can save it to the `transform.json` file. After that, this augmentation pipeline can be loaded from the `transform.json` file using the snippet above. The original `transform` and `loaded_transform` will contain exactly the same `Compose` object.

```
loaded_transform = A.load('/tmp/transform.json')
```

Augmentation pipelines are stochastic since each image transform operation is applied each time with some probability. During the debugging process, we may need to recover those exact

randomized parameters that were used to apply to a particular target. Replay mode implemented in Albumentations supports this functionality and it aims to improve reproducibility and developer experience while using the library.

### 4.6. Custom Augmentations

It is possible to extend augmentations without adding new classes using `A.Lambda` transformation. This class allows specifying a user-defined function for a chosen target. It allows easily extending Albumentations and reusing existing implementations of image processing routines without introducing changes to them. Let us consider a small, yet very illustrative example of the custom augmentation that imprints *A* logo on top of the image and the mask target (Figure 5).

```python
def custom_aug_img(image, **kwargs):
image_orig = image.copy()
cv2.putText(image, "A", ...)
cv2.circle(image, ...)
return cv2.addWeighted(image_orig, 0.5, image, 0.5, 0)

def custom_aug_mask(mask, **kwargs):
cv2.putText(mask, "A", ...)
cv2.circle(mask, ...)
return mask

custom_aug = A.Lambda(image=custom_aug_img, mask=custom_aug_mask)]
```

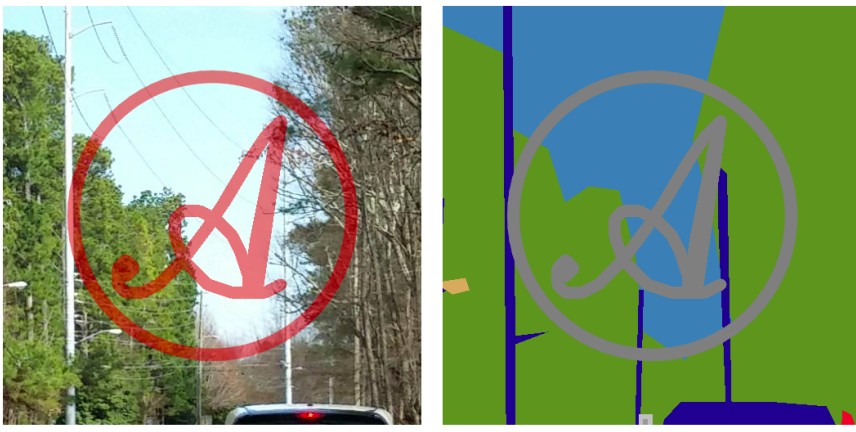

**Figure 5.** An example of applying a custom augmentation using `A.lambda` operator to an image (**left**) and a corresponding segmentation mask (**right**).

A more detailed code listing for this example is provided in the Supplementary Materials (Listing S12).

### 4.7. Performance

Albumentations follows the example of other popular Python machine learning packages and tends to minimize the use of pure Python functionality under the hood due to performance considerations. For example, vectorized functions are used as much as possible instead of loops. Furthermore, Albumentations considers multiple options for how each operation could be realized.

For example, Albumentations tries to work with images of `uint8` data type when possible for a number of reasons. First, it allows minimizing the memory usage and fitting more values into a SIMD register (e.g., $16 \times$ `uint8` vs. $4 \times$ `float32` values). Second, high-performance implementations

of common transform operations on `uint8` data are widely available. In some cases, it is possible for Albumentations to not perform transformations directly on the image, instead operating on the corresponding look-up table (LUT) and then applying it to the original image.

Although there are dedicated computer vision libraries such as OpenCV [17], which are also implemented in C++ and provide Python interface, their implementation of image transforms are not always the most efficient. For example, NumPy [64] implementation of an image flip operation used in Albumentations is faster than OpenCV implementation. The use of `numpy.where()` operator for conditional selection, `numpy.empty()` for memory pre-allocation, and `inplace` flag in supported NumPy operations can result in the sensible gains in the processing speed. To balance the performance and the number of underlying dependencies, we rely on a few low-level libraries that provide fastest implementations of almost all common image transforms, as demonstrated in Section 5.1.

## 5. Evaluation

### 5.1. Benchmarks

The quantitative comparison of image transformation speed performance for Albumentations and other commonly used image augmentation tools is presented in Table 1. We included a framework-agnostic image augmentation libraries imgaug [65], Augmentor [21], and SOLT [56], as well as augmentations provided with Keras [12] and PyTorch [66] frameworks. For the most image operations, Albumentations is consistently faster than all alternatives. Detailed instructions for running benchmarks locally are provided in the Albumentations GitHub repository: https://github.com/albumentations-team/albumentations.

**Table 1.** Results for running the benchmark on the first 2000 images from the ImageNet validation set using an Intel Xeon Platinum 8168 CPU. All outputs are converted to a contiguous NumPy array with the `np.uint8` data type. The table shows how many images per second can be processed on a single core (higher is better). **A** denotes results for Albumentations.

|  | A 0.4.2 | Imgaug 0.3.0 | Torchvision 0.4.1 | Keras 2.3.1 | Augmentor 0.2.6 | Solt 0.1.8 |
|---|---|---|---|---|---|---|
| HorizontalFlip | **2183** | 1403 | 1757 | 1068 | 1779 | 1031 |
| VerticalFlip | **4217** | 2334 | 1538 | 4196 | 1541 | 3820 |
| Rotate | **456** | 368 | 163 | 32 | 60 | 116 |
| ShiftScaleRotate | **800** | 549 | 146 | 34 | - | - |
| Brightness | **2209** | 1288 | 405 | 211 | 403 | 2070 |
| Contrast | **2215** | 1387 | 338 | - | 337 | 2073 |
| BrightnessContrast | **2208** | 740 | 193 | - | 193 | 1060 |
| ShiftRGB | **2214** | 1303 | - | 407 | - | - |
| ShiftHSV | **468** | 443 | 61 | - | - | 144 |
| Gamma | **2281** | - | 730 | - | - | 925 |
| Grayscale | **5019** | 436 | 788 | - | 1451 | 4191 |
| RandomCrop64 | **173,877** | 3340 | 43,792 | - | 36,869 | 36,178 |
| PadToSize512 | **2906** | - | 553 | - | - | 2711 |
| Resize512 | 663 | 506 | **968** | - | 954 | 673 |
| RandomSizedCrop64_512 | **2565** | 933 | 1395 | - | 1353 | 2360 |
| Equalize | **759** | 457 | - | - | 684 | - |

### 5.2. Ablation Study

To further experimentally demonstrate the efficiency of image augmentations, we conducted an ablation experiment in the binary image segmentation setting using the Inria Aerial Image Labeling dataset [62]. This dataset consists of 360 5000 × 5000 satellite images with annotated buildings in the form of binary masks of the same size. Exemplar images are shown in Figure 3. The ablation study was conducted as follows: We ran four different training sessions, of the same CNN architecture, using a fixed set of hyperparameters and changed only the level of image augmentations to evaluate their impact on the model performance on a binary image segmentation problem. The code listing for this experiment is provided in the Supplementary Materials (Listing S13).

We used the same metric as used on the official online evaluation server that is Intersection over Union (IoU) per image, averaged across all images. We report the best achieved IoU on the validation set for each augmentation level and data preprocessing time below. For this study, we used well-known UNet-based model architecture [67], with HRNetV2 encoder [68]. We did not use any pre-trained weights and started training from default initialization with a fixed random seed. Training ran for 100 epochs and RAdam optimizer [69] was used with the starting learning rate of $10^{-3}$ and cosine annealing to $10^{-6}$. Models were trained in a hardware setup with four 1080Ti NVidia GPUs with a batch size of 48 using PyTorch 1.4 [13] and NVidia Apex for mixed-precision training. Each training run took approximately 24 h. We used the Catalyst library [70] as a high-level framework for model training and experiment management.

Since original images are too large to fit in a GPU memory entirely, we randomly cropped a square image patch of the size from the range $[384; 640]$ from the source image and resized it to $512 \times 512$ during training. Other image augmentations were performed on the resized image. This cropping scheme was used in all of the following experiments:

1. No augmentations: After cropping and tile, no changes to the image were made.
2. Light augmentations: Random horizontal flips, change of brightness, contrast, color, and random affine and perspective changes.
3. Medium augmentations, an extended set of augmentations in addition to the Light scenario: Gaussian blur, sharpening, coarse dropout, removal of some buildings, and randomly generated fog.
4. Hard augmentations, extending the Medium set with: Random rotation by 90 degrees, image grid shuffle, elastic transformations, gamma adjustments, and contrast-limited adaptive histogram equalization.

The results for all four experiments are presented in Table 2. Without enough image augmentations, even our average-sized model (35M parameters) showed signs of overfitting after epoch 50, when IoU validation stopped improving. With Medium augmentations, the same model had a smaller gap between train and validation IoU scores and the best IoU was achieved towards the end of the training process, which shows the potential for even further improvement. When training with Hard augmentations, the model achieved the overall best validation IoU and did not overfit. The current state-of-the-art result on this dataset is 80.32 mIoU, and model trained with Hard augmentation have not reached this mark on the training set, which indicates it is still under-trained. Overall time for model training did not increase substantially, meaning that even Hard augmentations were fast enough to process the batch in time for passing it to a GPU.

**Table 2.** Results for different augmentation levels for the segmentation task on the Inria Aerial Image Labeling dataset [62]. Train IoU and Valid IoU show the best metric value reached across 100 epochs of training (higher is better). Data time and Model time indicate how long it takes to preprocess a batch of images and then run it through the network (lower is better).

| Augmentations | Train IoU | Valid IoU | Best Epoch | Data Time (sec/batch) | Model Time (sec/batch) |
|---|---|---|---|---|---|
| None | 84.67 | 73.89 | 45/100 | 0.09 | 0.6 |
| Light | 84.84 | 77.50 | 90/100 | 0.09 | 0.6 |
| Medium | 83.52 | 76.94 | 96/100 | 0.11 | 0.6 |
| Heavy | 79.78 | 78.34 | 95/100 | 0.13 | 0.6 |

This use case demonstrates that having the variety of different available transforms is important for preventing overfitting and achieving the best model performance. At the same time, thanks to the optimized operation performance in Albumentations, augmentation pipeline can be extended even further without slowing down training process.

### 5.3. Visualization

To anyone who works with image analysis, the ability to visualize the effect of programmatic operations applied to an image is of immense help. Qualitative visual inspection allows quickly validating the results of a transform and catch possible bugs early, especially in the case of the complex processing workflow. Thanks to the great community, there are two tools to visualize Albumentations (see Figure 6). The first one allows looking at the result of one specific transformation applied to one of the predefined images, manually change parameters to achieve the desirable result, and extract the exact Python code to reproduce it [71]. The second visualizes the output of a chain of transforms applied to a predefined or uploaded image in order to validate the resulting image [72]. Besides the obvious practical utility, both tools show involvement of the community around Albumentations.

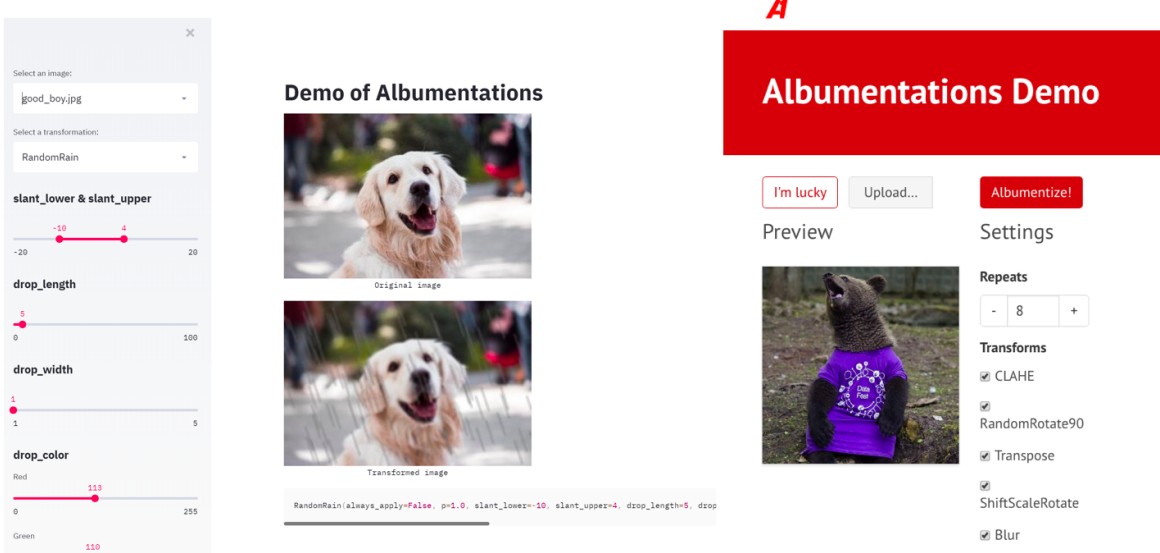

**Figure 6.** Community-developed tools to visualize the results of image transforms implemented in Albumentations: (**left**) visualization of a single transform with the ability for parameter tuning [71]; and (**right**) visualization of a chained number of transforms [72].

### 5.4. Adoption

Although there is no easy way to measure the adoption of the library, we can look at different metrics that may serve as a useful proxy. First, for any open source library, the number of stars on GitHub can show the interest of users. In Figure 7, we show the dependence of this metric as a function of time, as well as number of downloads via PyPI. Second, the library was born from winning solutions to the Computer Vision competitions, and it is not surprising that many, if not all, top teams at Kaggle use the library in their solutions [55]. Third, the library is gaining use in academia, as shown by recent Google Scholar mentions, with most common applications in biomedical and satellite image analysis. [73–80]. Finally, Albumentations has joined other PyTorch-friendly tools in the Pytorch ecosystem [81].

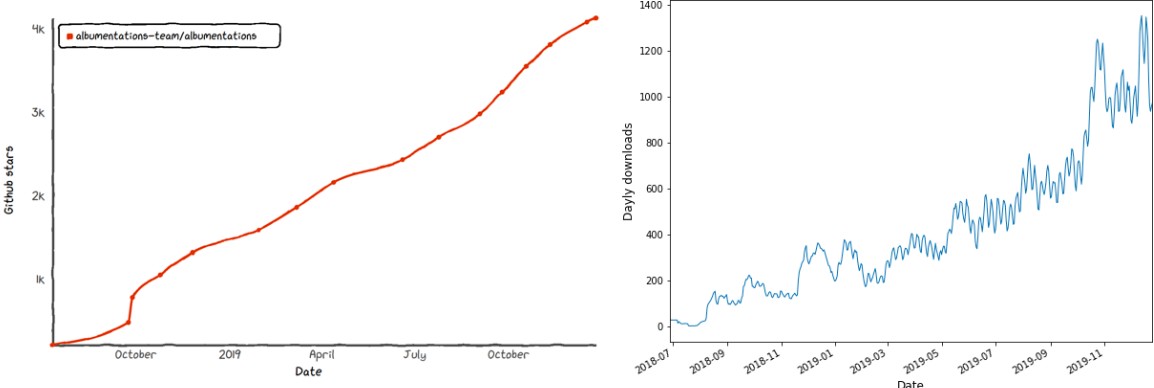

**Figure 7.** Library adoption shown as: (**left**) the number of stars in the Albumentations GitHub repository over time; and (**right**) the number of daily installations of the library using PyPI: *pip install albumentations.*

## 6. Discussion and Future Work

Augmentations proved to be a powerful approach that improves generalization and robustness of deep learning models. It is an active research direction, and the research community is coming up with new ways to use image augmentations and establish a solid theoretical framework behind their effects. Albumentations aims to balance among a few requirements, providing superior performance on a wide variety of transforms, coupled with the succinct API and an extendable structure. Quick adoption by the Kaggle, academic, and other communities validates the decisions made during the development of the library and provides invaluable feedback for the direction of future improvements.

As next step in the library development, we are exploring an option to be able to run augmentations on GPU. This is not a commonly used approach yet, but, according to recent reports, being able to use GPUs for augmentations may improve the training performance when the ratio GPU/CPU is high [43]. Another direction that we are exploring is to extend the transforms that Albumentations supports to 3D. Deep learning application in autonomous driving is a growing field and many tasks in the mapping and perception areas rely on LiDAR data. We believe that spatial transforms that work in 2D could be successfully applied to the 3D data.

**Supplementary Materials:** The following are available at http://www.mdpi.com/2078-2489/11/2/125/s1, Document: Albumentations Supplementary Materials.

**Author Contributions:** Software, A.B., A.P., E.K., V.I.I., and M.D.; validation, A.B., A.P., E.K., V.I.I., M.D., and A.A.K.; writing—original draft preparation, E.K., V.I.I., and A.A.K.; writing—review and editing, A.B., A.P., E.K., V.I.I., M.D., and A.A.K.; and visualization, E.K. and V.I.I. All authors have read and agreed to the published version of the manuscript.

**Funding:** This research received no external funding.

**Acknowledgments:** We are grateful to all GitHub contributors and those who reported bugs and provided constructive feedback. We also thank the Open Data Science (ODS.ai) community [82] for useful suggestions and other help aiding the development of this work. A.K.K. thanks Xin Rong of the University of Michigan for the donation of the Titan X NVIDIA GPU.

**Conflicts of Interest:** The authors declare no conflict of interest.

## Abbreviations

The following abbreviations are used in this manuscript:

| | |
|---|---|
| API | Application Programming Interface |
| CPU | Central Processing Unit |
| GPU | Graphics Processing Unit |
| IoU | Intersection over Union |
| JSON | JavaScript Object Notation |
| LUT | Look-Up Table |
| PCA | Principal Component Analysis |
| PyPI | Python Package Index |
| RGB(A) | Red, Green, and Blue (Alpha) |
| SIMD | Single Instruction, Multiple Data |
| YAML | YAML Ain't Markup Language |

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
