# Peer review of "Albumentations: Fast and Flexible Image Augmentations"

_information, doi:10.3390/info11020125_

Round 1

Reviewer 1 Report

1- on line 33, the authors mention "this approach could be challenging, time-consuming, and error-prone". I am not sure exactly which approach they are talking about since they had mentioned several methods in the previous sentence.

Moreover, the authors should give evidence on the claim that it is challenging, time-consuming and error-prone. 

2- It is recommended to add a flowchart of the designed Albumentations method. 

Author Response

We are thankful to the reviewer for the constructive feedback. We have carefully reviewed the comments and revised the manuscript accordingly. Our point-by-point responses are given below.

> 1- on line 33, the authors mention "this approach could be challenging, time-consuming, and error-prone". I am not sure exactly which approach they are talking about since they had mentioned several methods in the previous sentence.

Moreover, the authors should give evidence on the claim that it is challenging, time-consuming and error-prone. 

Response: This claim was referring to the situation in which complex image augmentation algorithms are re-implemented from scratch, which takes more time and can lead to programming errors. We clarified the corresponding sentence in the manuscript (lines 34-37), provided the reference to the Nature paper supporting this claim and also referred to the further discussion in the section 2.3 on Augmentations of complex targets.

> 2- It is recommended to add a flowchart of the designed Albumentations method. 

Response: We added two UML diagrams describing the structure of Albumnetations library to Supplementary Materials. The first shows the class hierarchy of different transforms, and the second describes classes of transform compositions.

Reviewer 2 Report

Although the work of the authors is important and benefits the deep learning community, this work is not research in my opinion.

Experiences should be more in depth: test on different computer configurations (multi-cores) and show the interest of this library on use cases and not just on 2000 random images.

I would have liked to see experiments around the cases of segmentation and detection which seem a characteristic of this library. In particular, the gain in time and accuracy (ghost pixels removed) that this new library allows?

Likewise quantify the gain of the visualization tools which seem very interesting. How does this impact learning?

My main concern is that in general the manuscript looks more like computer documentation than a research article. Many of the descriptions of the use of the library should appear on the GitHib or as supplementary materiel and not in a research manuscript.

Last point, I'm not sure I understand how the time associated with the increase in data is so critical. It seems to me that this is done before learning and there is therefore no real competition with learning in terms of resources. The authors should be clearer on this point.

Author Response

We are thankful to the reviewer for the invaluable and constructive feedback. We have carefully reviewed the comments and revised the manuscript accordingly. Our point-by-point responses are given below.

1. Although the work of the authors is important and benefits the deep learning community, this work is not research in my opinion.

> This manuscript is different from most publications in the ML/DK community, as it is more of the system design paper. However, we would like to argue that our contributions are novel and important:

To outline the motivation for building this library, in Introduction we first reviewed how and where image augmentations are used and identified 3 main challenges (or needs) that currently exist, Based on that we formulated system design principles that underlie our approach to addressing these challenges and discussed some trade-offs for the design decisions that we’ve made, We described key features of the library that implement the proposed design and showed through examples how they can be used to satisfy the identified needs. Through benchmarks, adoption examples, and a new use case, we demonstrate the benefits of using the library in practice in both research and industry.

We have not seen such an analysis for any of image augmentation tools in the previously published literature. Our contributions can be considered higher-level compared to some other system design papers (e.g. on TensorFlow or PyTorch), however, that does not necessarily discount their research value. Finally, we think this type of system paper is a good fit for the scope of this specific special feature of Information, as it is directly related to using the tool in the Python ecosystem to help solving important Machine Learning problems.

We made multiple changes in the text of the manuscript to improve it according to the suggested recommendations, please see other points below:

2. Experiences should be more in depth: test on different computer configurations (multi-cores) and show the interest of this library on use cases and not just on 2000 random images.

> Low-level image operation libraries (e.g. NumPy or OpenCV) can use multi-threaded operations. However, most deep learning frameworks (e.g. PyTorch or TensorFlow) implement their own parallel processing to apply transformations to a number of input images (a batch). We do not mix both levels of parallelism as this may lead to hard-to-catch bugs or deadlocks. Instead, we rely on the framework’s implementation of multiprocessing as recommended by their Best practices (e.g. see https://pytorch.org/docs/stable/notes/multiprocessing.html). Multiprocessing allows to independently read and pre-process each image separately and, thus, provides a near-linear speedup. Therefore, multiprocessing capability is independent of our library, while Albumentations focuses on the performance of processing each single image, as reflected in benchmarks. We did update text in section “2.2. Performance considerations” to clarify the use of multiprocessing. We also did add a new use case with some performance measures using multiprocessing, please see the next point.

3. I would have liked to see experiments around the cases of segmentation and detection which seem a characteristic of this library. In particular, the gain in time and accuracy (ghost pixels removed) that this new library allows?

> Per reviewer’s suggestion, we added a new extensive ablation study experiment for the image segmentation problem, in which we compared 4 different levels of augmentation–please see the section “5.2 Ablation study”. There we showed that using more sophisticated image transformations allows boosting prediction accuracy and avoid overfitting, without increasing the overall time for training a model due to the high image processing speed of Albumentations. Thus, optimized augmentations allow sustaining high GPU utilization, such that one can add more rich and sophisticated augmentations scenarios to further diversify the dataset while not increasing the time needed for 1 epoch of training.

4. Likewise quantify the gain of the visualization tools which seem very interesting. How does this impact learning?

> Visualization tools were developed by the community and serve rather as convenience tools that allow to interactively preview the effect of an image transform(s) with the specific set of parameters on a chosen image. These tools do not impact model learning process directly, however, they can be used as a sanity check to empirically choose the acceptable range of image augmentation parameters such that semantic information useful for the training is not lost after transformations. There have been recent developments in automatically learning these sets of parameters for a specific dataset–we mention some of those in the Introduction, including AutoAugement and Population Based Augmentation.

5. My main concern is that in general the manuscript looks more like computer documentation than a research article. Many of the descriptions of the use of the library should appear on the GitHib or as supplementary materiel and not in a research manuscript.

> We agree, that section “4. Key features” was saturated with too many technical details. Therefore, we moved full detailed code listings to the Supplementary material and left only parts that are necessary for demonstrating how the main capabilities of Albumentations implement the proposed design.

6. Last point, I'm not sure I understand how the time associated with the increase in data is so critical. It seems to me that this is done before learning and there is therefore no real competition with learning in terms of resources. The authors should be clearer on this point.

> It is a common practice that image augmentations are not pre-computed, but rather performed during training on-the-fly, as implemented in all major deep learning frameworks. This allows one to overcome limitations of the storage space since the datasets are typically quite large without sacrificing the diversity of the transformed images. For example, in the use case we just added, we trained 4 models for 100 epochs each on the Inria training set of 13.3GB. If we pre-computed all augmentations, it would take 5.2TB of storage space. Moreover, on-the-fly operations also allow quickly iterate during prototyping, debugging and modifying the protocol, as there is no need to again re-compute all the stored augmented images. We updated section “2.2. Performance considerations” to clarify that.

Reviewer 3 Report

This article describes a library for image processing with some image transform operations among other things. The authors show some examples.

In some cases, the authors should describe some acronyms the first time they are mentioned in the text.

I find the paper to be interesting, the design principles are well planned and well described. In general the content of paper is well described. The contribution seems specified in the first parts and I consider it a good contribution to the journal.

I agree with the authors, the next step in the library development should be exploring the option with GPU.

Author Response

We are thankful to the reviewer for the constructive feedback. We have carefully reviewed the comments and revised the manuscript accordingly.

> In some cases, the authors should describe some acronyms the first time they are mentioned in the text.

Response: We made sure to introduce abbreviations after the first use of the full term, such as "application programming interface (API)" (lines 39-40), "graphics processing units (GPUs)" (lines 85-86), etc.

Round 2

Reviewer 2 Report

I really enjoyed reading the authors' responses and their real desire to take my comments into account.
The manuscript is much clearer on the details I was missing.
Likewise, I appreciate the ablation study experiment for the image segmentation problem that has been added. This convinces me really of the usefulness of the library for image augmentation.

Author Response

We are thankful to the reviewer for the constructive feedback that helped to improve the manuscript.